# Genomic Characterization and Wetland Occurrence of a Novel *Campylobacter* Isolate from Canada Geese

**DOI:** 10.3390/microorganisms11030648

**Published:** 2023-03-03

**Authors:** David M. Linz, Kyle D. McIntosh, Ian Struewing, Sara Klemm, Brian R. McMinn, Richard A. Haugland, Eric N. Villegas, Jingrang Lu

**Affiliations:** 1Oak Ridge Institute for Science and Education, Oakridge, TN 37830, USA; 2Office of Research and Development, United States Environmental Protection Agency, Cincinnati, OH 45268, USA; 3EPA National Student Services Contact (NSSC), Cincinnati, OH 45268, USA

**Keywords:** *Campylobacter canadensis*, Canada goose, genome sequence, microbial source tracking, wetland

## Abstract

Populations of resident, non-migratory Canada geese are rapidly increasing. Canada geese are known to transmit viral and bacterial diseases, posing a possible threat to human health. The most prevalent pathogens vectored by geese are *Campylobacter* species, yet the current understanding of the identity and virulence of these pathogens is limited. In our previous study, we observed a high prevalence of *Campylobacter* spp. in the Banklick Creek wetland—a constructed treatment wetland (CTW) located in northern KY (USA) used to understand sources of fecal contamination originating from humans and waterfowl frequenting the area. To identify the types of *Campylobacter* spp. found contaminating the CTW, we performed genetic analyses of *Campylobacter* 16s ribosomal RNA amplified from CTW water samples and collected fecal material from birds frequenting those areas. Our results showed a high occurrence of a *Campylobacter canadensis*-like clade from the sampling sites. Whole-genome sequence analyses of an isolate from Canada goose fecal material, called MG1, were used to confirm the identity of the CTW isolates. Further, we examined the phylogenomic position, virulence gene content, and antimicrobial resistance gene profile of MG1. Lastly, we developed an MG1-specific real-time PCR assay and confirmed the presence of MG1 in Canada goose fecal samples surrounding the CTW. Our findings reveal that the Canada goose-vectored *Campylobacter* sp. MG1 is a novel isolate compared to *C. canadensis* that possesses possible zoonotic potential, which may be of human health concern.

## 1. Introduction

From 1970 to 2012, resident non-migratory Canada geese (*Branta canadensis*) populations increased from 1.26 million to 5.69 million in North America [1]. While this increase may support recreational activities such as bird watching, high goose occupancy can lead to undesirable consequences such as agricultural damage, accumulation of fecal material on surfaces, and degraded water quality [2]. Waterfowl such as Canada geese are known to transmit viral and bacterial diseases of human and agricultural concern [3,4]. Their feces are a source of pathogens such as avian influenza, *Arcobacter*, *Escherichia coli*, *Salmonella*, *Giardia, Cryptosporidium*, and *Campylobacter,* which can be released into freshwater reservoirs [3,5,6,7,8]. Among those potential pathogens, *Campylobacter* spp. can be highly prevalent (11.2%) in Canada geese, suggesting that these birds may play a role in dispersing *Campylobacter* within their frequented environments [9].

Any environment frequented by wild birds, and waterfowl specifically, could contain a diverse array of *Campylobacter* spp. (for review see [10]). Sanitation District No. 1 (SD1) in Fort Wright, Kentucky (USA) installed a constructed treatment wetland (CTW) in 2011 to divert and treat a portion of the Banklick Creek, a watershed regularly impacted by combined sewer overflows (CSOs) and sanitary sewer overflows (SSOs) in the area. CTWs can be an effective means to passively treat water by removing harmful chemical and microbial contaminates, but since they closely mimic natural wetlands, they can attract native wildlife such as waterfowl. Previous research using qPCR analyses of the treated water throughout the system showed that *Campylobacter* and avian marker signals (targeting *Helobacter* sp.) were detected at high frequencies where bird occupancy was commonly observed [11]. Moreover, various waterfowl species, such as Canada goose, were frequently found throughout the CTW. However, more in-depth characterization of the *Campylobacter* spp. diversity detected in this wetland, as well as establishing if Canada geese could be vectors of human pathogenic *Campylobacter* strains in these environments were not conducted. To understand the molecular basis of pathogenicity, the identification of virulence factors is needed to elucidate the potential of a *Campylobacter* isolate to cause disease [12]. For those well-known *Campylobacter* species (*C. jejuni* and *C. coli*) which cause human (and animal) infections, several putative virulence factors (genes) that contribute to motility, intestinal adhesion, colonization, toxin production, and invasion have been identified as critical to their human and animal pathogenicity. For example, the genes *cdtA*, *cdtB*, and *cdtC* [13,14] encode *Campylobacter* cytolethal distending toxin, causing host cell cycle arrest, cell distention, and eventually cell death [15]. Some genes such as *flaA* encoding flagellin [16] and *cadF* encoding a protein that interacts with a host extracellular matrix protein fibronectin [17] are required for *Campylobacter* adherence to, and colonization of the host cell surface. Although *Campylobacter* spp. have been isolated from Canada geese as mentioned above, documentation addressing their human pathogenicity is rare, and data on the virulence of *Campylobacter* strains isolated from the large resident populations of Canada geese in urban and suburban settings are limited.

In this study, we began by using genetic analysis to parse the possible identity of targeted *Campylobacter* 16s rRNA genes amplified from the CTW. We found that a large portion of the sequenced clones originated from a *Campylobacter canadensis*-like species of unknown identity. Next, we tried to isolate the potential *Campylobacter* spp. from CTW Canada goose fecal samples, which failed despite multiple attempts. Assuming the age of the fecal samples was hampering our efforts, we collected fresh fecal samples from a Mason city park (40 miles away from the CTW), located near our laboratories, where we could monitor goose activity and collect droppings immediately. From the Mason samples, we identified a unique isolate, which we called MG1. Initial 16s rRNA sequence analysis showed high similarity between MG1 and isolates from the CTW. To further understand MG1 at a genetic level we performed genome sequencing and assembled an annotated draft genome. We used phylogenomics to explore the relationship of the MG1 isolate to other *Campylobacter* species and confirmed the identity of the unknown clone sequences from the CTW as most closely related to MG1. We also surveyed the MG1 genome for loci associated with virulence and disease in humans to begin to understand the zoonotic potential of the isolate. Finally, we designed a unique real-time PCR assay for detecting and tracking MG1 in goose fecal contamination in CTW water samples. Our findings help to understand host-specific strains of *Campylobacter* spp. and aid in microbiological water source tracking. These data contribute to an overall effort to safeguard public water ecosystems and will assist in preventing future human and animal exposure.

## 2. Materials and Methods

### 2.1. Fecal Sample Collection, Isolation and DNA Extraction

Initially, two collections of Canada goose fecal material with an unknown time of defecation were collected from the CTW (Ft. Wright, KY, USA) in fall 2017. After failing to isolate *Campylobacter* spp. from these samples, fresh Canada goose fecal samples were collected from local, easily accessible city park lawns where geese are frequently found and monitored (Mason, OH, USA) during the following year. Fecal samples were collected just after defecation, immediately transferred to a sterile 50 mL conical tube (ThermoFisher Scientific, Waltham, MA, USA), and brought to the US EPA laboratory in Cincinnati, OH for further processing. Each fecal pellet was submerged in sterile 1× PBS solution and vortexed to homogenize the sample. Duplicate aliquots (100 µL) were diluted 10-fold in sterile PBS and cultured as previously described [18]. Briefly, each sample was spread on two 47-mm diameter polycarbonate filters of 0.6-µm pore size (GE Water & Process Technologies, Addison, IL, USA) placed on tryptic soy agar TSA *w*/5% Sheep’s Blood (BD biosciences, San Jose, CA, USA). The plates were incubated at 37 °C, which was found to be the optimal temperature for the recovery and growth of most Campylobacter species [19], for 30 min. Following incubation, the filters were removed, and the plates were grown in a microaerophilic chamber at 37 °C for 48 h. Presumptive *Campylobacter*-positive colonies were picked and streaked on TSA *w*/5% Sheep’s Blood plates and grown at 37 °C in a microaerophilic chamber for 48 h for isolation. From the plates, a single colony was selected for DNA extraction using the Tissue and Cell Lysis kit (Epicenter Technologies Corp. Madison, WI, USA) following the manufacturer’s instructions. DNA was screened for *Campylobacter* using the 16s PCR assay described by Linton et al. 1996 [20]. The isolates that were positive by PCR were stored at −80 °C in tryptic soy yeast broth (TSY) containing 15% glycerol.

### 2.2. Illumina MiSeq Genomic Sequencing, Read Processing, and Assembly/Annotation

From positive *Campylobacter* isolates (see above), a sequencing library was prepared using Nextera XT (Illumina) following the manufacturer’s protocol. The libraries were denatured and diluted following the MiSeq Denature and Dilute library guide to a final concentration of 10 pM and mixed with 10% PhiX sequencing control. The pooled library was then loaded into a MiSeq Reagent Kit v3 and run using 300 base-pair (bp) paired-end chemistry (Illumina, San Diego, CA, USA). During this effort, 3,722,434 paired reads were produced for further analysis. Raw reads with primers and adapters removed were then processed. Reads were quality checked using Fastqc v.0.11.9 [21] and cleaned using Trimmomatic v.0.39 [22] to perform quality trimming. Trimmomatic was run with the following parameters, LEADING:2 TRAILING:20 SLIDINGWINDOW:5:20 MINLEN:150. After trimming, Fastqc was run again. Approximately 9% of reads were removed by trimming. The genome was assembled with the SPAdes assembler v3.11.1 [23] using default parameters, and whole-genome annotation was performed with PROKKA v1.14.6 [24]. Assembly quality was evaluated with QUAST v5.0.2 [25]. Genome assembly information is presented in Table 1.

### 2.3. Antibiotic Resistance Gene, Virulence Gene, Lipooligosaccharide Biosynthesis Loci, and Capsular Polysaccharide Analyses

To check for antimicrobial resistance genes, we used the Comprehensive Antibiotic Resistance Database (CARD, version 3.2.3) RGI tool (version 5.2.1) that predicts Open Reading Frames (ORFs) using Prodigal, homolog detection using DIAMOND, and Strict significance based on CARD curated bitscore cutoffs [26]. For analysis of virulence genes, lipooligosaccharide (LOS), and capsular polysaccharide (CPS) regions, we used BLAST v2.9.0+ [27]. Custom BLAST databases were generated from the PROKKA amino acid annotations from the MG1 isolate genome. Query virulence gene sequences were taken from *Campylobacter jejuni* NTCT11168 or other species where appropriate. For all BLAST analyses, alignments were inspected by hand, and e-value cutoffs were made at 1 × 10^−5^. Reciprocal BLAST searches were performed to confirm orthology when positive hits were detected.

### 2.4. Water Sampling, PCR, Cloning, and Sanger Sequencing for Banklick Creek Wetland Samples

The wetland sampling has been described in our previous study [11]. Briefly, samples were collected from June through September 2017 from five sites in the Banklick Creek Treatment Wetland located in Fort Wright, Kentucky (39°01′14.2″ N 84°31′42.1″ W) (Appendix A). Water samples (~10 L) were collected in a sterilized carboy from each of the sampling locations over a 1-h long sampling event, and immediately transported to the U.S. EPA laboratory located in Cincinnati, Ohio for processing. Bird occupancy was also recorded at each of the five sampling locations during each of the 15 independent sampling events. Recorded bird occupancy was then totaled for the entirety of this study at each sample site. For the collected water samples, duplicate 200 mL aliquots from each 10 L carboy were passed through a 47 mm polyvinylidene difluoride (PVDF) filter with a nominal pore size of 0.45 μm (Millipore, Burlington, MA, USA) (Total: 15 × 5 × 2). Each membrane was transferred to a Lysing Matrix A bead tube (MP Biomedicals, Santa Ana, CA, USA), and the resulting filter tubes were stored at −80 °C until extraction. To each filter tube, 600 μL RLT Plus buffer was added and the tube underwent bead beating (5000 reciprocations/min) for 30 s, cooling on ice for 5 min, exposure to another round of bead beating, and centrifugation for 5 min at 12,000 RPM. Nucleic acids were then extracted from the supernatant using the AllPrep DNA/RNA Mini Kit (Qiagen, Germantown, MD, USA), following the manufacturer’s instructions. A *Campylobacter* genus-specific PCR assay was used to amplify 28 of the samples which were positive via the previous qPCR assay for *Campylobacter* genera [11,20,28] (Appendix A). The amplicons were cloned into pCR4.1 TOPO (Invitrogen, Carlsbad, CA, USA) to aid in further identification. Raw sequences were processed with Sequencher 5.2.4 software (Gene Codes, Ann Arbor, MI, USA) for editing, initial comparisons, alignment, homology searches, and a chimera check as previously described [29].

### 2.5. 16s rRNA Phylogenetics and Phylogenomic Analyses

To resolve the phylogenomic position of *Campylobacter* isolate MG1, we utilized GToTree v1.6.11 [30] with a variety of *Campylobacter* genomes from GenBank. GToTree automates querying and downloading all genomes from GenBank, translating the MG1 isolate genome assembly in all open reading frames, filtering all genomes for 119 possible single-copy genes against the Proteobacteria HMM-gene set, retaining only single-copy genes present in at least 90% of all genomes which, for the genomes we chose for analysis, retained ~108 genes, on average. GToTree then performs multiple sequence alignments followed by phylogenetic reconstruction of the concatenated multiple sequence alignment using heuristic neighbor-joining with the consensus tree calculated from 1000 bootstrap replicates. Average nucleotide identity (ANI) was used to contrast closely related genomes and assess species similarity [31,32].

To analyze the phylogenetic distribution of *Campylobacter* isolates detected at sites 3 and 5 in the Banklick Creek Wetland area (see Appendix A), we gathered the 16s rRNA nucleotide sequences (see above), together with nine additional 16s rRNA sequences from known *Campylobacter* spp., as well as the 16s sequence from the MG1 isolate, for a total of 38 nucleotide sequences. Sequences were compiled and aligned using MUSCLE in MEGA X [33] with default settings. Alignments were manually inspected and trimmed to remove gaps. The evolutionary history was inferred using the neighbor-joining method in MEGA X. The bootstrap consensus tree was inferred from 500 replicates. The evolutionary distances were computed using the Maximum Composite Likelihood method to estimate the base substitution rate. All positions containing gaps and missing data following alignment were eliminated (complete deletion option). In a total, 661 positions were included in the final dataset.

### 2.6. MG1 16s rRNA Primer Design

To design primers specific to MG1 16s rRNA, we obtained 16s rRNA nucleotide sequences from five *Campylobacter* species (*C. lari*—accession: CP046243.1, *C. coli*—accession: CP083814.1, *C. rectus*—accession: CP012543.1, *C. jejuni*—accession: CP048760.1, and *C. canadensis*—accession: CP035946.1) and performed multiple sequence alignment against the MG1 isolate 16s rRNA nucleotide sequence (Appendix A). We inspected the alignment and designed primers and a probe placed in regions with the greatest number of nucleotide differences. Using BLAST, we inspected the sequence region of each species where primers were designed to check for a lack of inter-species variation. Three sets of real-time PCR assays were designed, and in silica analysis was conducted for the oligo sequence of each set to ensure that at least one primer and the probe of each assay was specific to MG1. It is worth noting that, because of the sequence conservation, we were unable to locate a target region for MG1 where the primers were highly unique compared to *C. canadensis*; however, the forward primer sequence contained at least 1 unique residue, while the probe sequence contained 3 unique residues. Lastly, we evaluated the three assays in terms of their sensitivity and amplification efficiency and a single primer/probe set was selected and used for this study. The MG1 assay was also cross-tested against various environmental and laboratory *Campylobacter* strains and three clinical strains to assure no false-positive detection occurred [18]. For this analysis, real-time PCR was performed as described below (also see Appendix A).

### 2.7. Real-Time PCR Analysis

Real-time PCR assays were used to detect the MG1 *Campylobacter* isolate for samples from Canada geese excreta. Real-time PCR was performed on a QuantStudio 6 Flex thermal cycler using Applied Biosystems^®^ TaqMan Environmental Master Mix (EMM) 2.0 (Life Technologies, Grand Island, NY, USA). The real-time PCR assay was carried out in a 20 μL volume and contained 2 μL DNA, 10 μL 2× EMM, 3 μL of forward and reverse primers and probe mixture (100 nM), and 5 μL DNase-free water. The cycling conditions consisted of a 2 min hold at 50 °C, then a 10 min hold at 95 °C, followed by 40 cycles of 15 s at 95 °C and 1 min at 60 °C. The standards of *Campylobacter* spp. (MG1) were diluted to yield a series of 10-fold concentrations ranging from 10^1^ to 10^7^ copies, prepared in triplicates, and then used for standard curves.

## 3. Results

### 3.1. Bird Activities and Campylobacter Genetic Analysis from Banklick Creek Wetlands

At the Banklick Creek CTW (Appendix A), we observed that bird occupancies varied among the five sites with different vegetative covers. At sites 1 and 2 (no vegetative cover), negligible numbers of birds (i.e., 1) were observed only during one and two sampling events for these sites, respectively. At site 3 (most vegetative cover), birds were observed during 60% of the sampling events (ranging from 1 to 10 birds). Decreased bird occupancy was noted at site 4 (less vegetative cover) with ranges of 2–10 birds present in only 20% of sampling events. At site 5 (outlet), no bird occupancy was observed during the entirety of this study. The previously detected *Campylobacter* quantity showed a positive correlation with bird occupancies and avian marker qPCR signals [11].

We next analyzed the composition of the *Campylobacter* isolates cloned from sites 3 and 5 at the CTW. To assess the identity of the isolates present at the two sites and compare the taxonomic composition of isolates derived from each location, we cloned and sequenced the 16s rRNA region of 28 isolates from both sites and phylogenetically compared them to 16s sequences of various *Campylobacter* species (Figure 1). A majority of the 28 isolates clustered within a clade containing *C. canadensis*; however, the 16s sequences were, on average, only 94.4% identical between *C. canadensis* and the unknown clones. These data suggested that the clones may be a unique species compared to *C. canadensis* [34]. Accordingly, significant efforts were made to isolate *Campylobacter* spp. from the Canada goose fecal samples collected from the Banklick Creek CTW. The isolation effort initially targeted Canada goose fecal samples collected from different dates at the CTW, but this effort failed likely due to the age of feces collected. Instead, we acquired fresh Canada goose feces from local locations (e.g., Mason, Ohio) for our analysis.

### 3.2. Campylobacter MG1 Genome and MG1 Phylogenomic and Phylogenetic Analysis

After successfully isolating a novel *Campylobacter* isolate, termed MG1, from goose fecal samples (see materials and methods) we performed genome sequencing. The assembled MG1 genome was of good quality with high coverage, containing 1.9 Mb of genomic sequence composed of 92 contigs with an N50 length of 1.7 × 10^5^ bp. Our annotation efforts produced 1908 genes (Table 1). We used phylogenomics to better understand the phylogenetic relationship of MG1 to other *Campylobacter* species. The novel MG1 isolate clustered within a distinct clade that includes *C. canadensis* and a recently published *Campylobacter* isolate taken from a cloacal swab of a Canada goose from California (strain RM12654, accession: GCA_020137585.1) (Figure 2). To further understand the relationship between the clade containing *C. canadensis,* strain RM12654, and MG1 we directly aligned 16s rRNA sequences and performed an average nucleotide identity (ANI) analysis among the genomes. Compared to *C. canadensis* and RM12654 (GenBank: CP035946.1 and MW131451.1, respectively), the MG1 16s rRNA sequences were 96.8% and 100% similar, and the ANI values were 74.6% and 88.8%, respectively. The level of variation (3.2%) in the 16S rRNA sequences between the two clades (MG1 and *C. canadensis*) was significant according to [34]. Next, we repeated our phylogenetic analysis of the CTW clones and integrated the 16s rRNA sequence from MG1. The 20 clones of interest clustered within a clade containing the MG1 isolate and were on average 98.5% identical (97.7–98.9%) (Figure 1). The variations (1.2%) among the 20 clones suggested a reasonable intraspecies diversity according to previous analysis for *Campylobacter* species [34]. The nearest outgroup to this clade was *C. canadensis*—matching our initial analysis and the above phylogenomics (Figure 1 and Figure 2).

### 3.3. Virulence Gene Content and Antibiotic Resistome of the MG1 Strain

To determine the virulence gene content of the MG1 isolate, we assembled a list of potential virulence genes reported to contribute to the virulence of *Campylobacter* in the literature [35,36,37,38]. These included categories of virulence such as genes and gene clusters involved in adhesion, invasion, CPS synthesis, LOS synthesis, and type IV secretion (T4SS). We used amino acid sequences of these genes to query the amino acid sequences of annotated genes from the MG1 isolate using BLASTp.

Most categories of virulence factors were identified in the MG1 genome, including factors associated with adhesion and colonization (*cadF*, *jlpA*, *racR*, *racS*, and *peb* genes), invasion (*ciaB*, *iamA*, and some but not all flagella-associated genes), T4SS genes (albeit with weak orthology), and other factors. We did not detect toxin-producing genes in MG1 (*cdt* genes) (Table 2). Next, we explored genes associated with LOS and CPS. Specifically, we searched for the genes involved in sialic acid synthesis and translocation (*cst-III*, *neuB1*, *neuC1*, and *neuA1*) that define the group 1 category of the LOS locus, including types A, B, C, R, M, and V—the types often associated with neural disease development (including Guillain–Barré syndrome and Miller Fisher syndrome) [36]. MG1 possessed orthologs of all 4 LOS genes associated with group 1 LOS types (Table 3). We also explored the CPS loci in MG1 and found an intact CPS region containing ~19 CPS orthologs, 12 of which were syntenous across a singular contig. However, we cannot rule out the possibility that the fractionation of our genome assembly hindered our ability to detect synteny of the entire CPS locus (Table 4). Lastly, we checked for the presence of antimicrobial resistance (AMR) genes in the MG1 isolate but did not detect any AMR-associated genes.

### 3.4. MG1-Specific Detection in Canada Goose Fecal Samples

As a final step in our analysis, we designed primers and a probe specific to the MG1 isolate 16s rRNA (see materials and methods) and tested our assay against an array of *Campylobacter* species to confirm the primer/probe specificity (Appendix A). Once confirmed, we assessed the prevalence of the MG1 isolate directly from Canada goose fecal samples we collected from the areas just outside of sites 3 and 5 at Banklick Creek CTW. MG1 was detected in each of the samples we collected (Appendix A).

## 4. Discussion

### 4.1. Bird Activities and Campylobacter Genetic Analysis from Banklick Creek Wetlands

We found all but eight of the sequenced isolates clustered within a clade containing MG1. We then confirmed that these isolates were likely the same species based on 16s rRNA homology. This finding implies that MG1 is pervasive in the CTW where Canada geese frequent. The nearest outgroup to the MG1 clade was *C. canadensis*, a novel *Campylobacter* isolate obtained from cloacal swabs from captive adult whooping cranes (*Grus americana*) [39]. A small subset of three clones from site 3 formed a clade with *C. lanienae*, a natural inhabitant of healthy farm and feral animals [40,41,42,43]. The remaining five clones clustered with the additional *Campylobacter* spp. included in the phylogeny. The prevalence of Canada geese within the CTW suggested that geese were the predominant source of MG1 detected in the wetland area. In light of these data, future studies will focus on the prevalence of MG1 in other habitats that have high Canada goose occupancy.

### 4.2. Campylobacter MG1 Genome and MG1 Phylogenomic and Phylogenetic Analysis

Our initial phylogenomics suggested that MG1 is a novel or unique isolate separate from *C. canadensis*. These data also suggest that the MG1 isolate is likely the same species as the RM12654 strain [31,34]. This close relationship to the RM12654 strain from a Canada goose in California indicates that MG1 and MG1-related species may be present in Canada geese from across the United States. Nevertheless, a DNA sequence analysis of 16s rRNA genes for eight isolates from chicken carcasses collected from Saudi Arabia showed the highest identity to Campylobacter strain RM12654 16s ribosomal RNA gene [44], suggesting that this *Campylobacter* sp., originally isolated from Canada geese, is also able to reside in chickens, although it is currently unclear which animal could be the original host. The host lability of *Campylobacter* spp. and their ability to overlap between humans, domestic animals, and wild birds further highlights the public health concern generated by the ability of wild birds to mobilize these strains across large geographic distances [10].

Altogether, our data support our previous results describing *Campylobacter* spp. present at sites 3 and 5 [11]. Our data also show that the majority of the clones we sequenced from the sites were phylogenetically identified as most closely related to MG1. Unfortunately, our sampling scheme was not structured to target times of high goose occupancy. In addition to testing pervasiveness across environments, future studies should monitor how the abundance of MG1 changes in single sites within the CTW over time with the fluctuation in Canada geese occupancy.

### 4.3. Virulence Gene Content and Antibiotic Resistome of the MG1 Strain

Our analysis of virulence gene content in MG1 showed that the isolate contains many genes associated with virulent tendencies among *Campylobacter* spp. The presence of LOS group 1 and CPS loci indicate that this isolate can trigger neural diseases often associated with other *Campylobacter* spp.—although the direct association with these loci and disease is an active area of research [36]. Despite these findings, the percent amino acid similarity we detected was generally low, although this is likely a product of the evolutionary distance between our query sequences (predominantly *C. jejuni*) and the MG1 isolate. Further, we failed to identify *cdt* genes within MG1 which, when absent in *C. jejuni*, was shown previously to lessen the severity of infections [45]. We did not detect isolates similar to MG1 derived from human patients infected with *Campylobacter*. Combined, this may suggest that MG1 poses minimal infectious risk to humans, and the virulence genes we detected generally function to facilitate infection in avian hosts. Despite this, the genetic potential for future human-related virulence cannot be ruled out. The absence of AMR and toxin-producing genes present in MG1 lessens the danger of human and animal transmission or health effects should MG1 be found in non-avian hosts in the future.

The lack of AMR genes is a promising finding given the increasing prevalence of resistance to common antibiotics found among *Campylobacter* spp. frequently found in wild birds [10,46]. These findings may imply a lack of significant crossover between MG1, MG1 avian vectors, and human/human-related waste, as AMR prevalence is associated with the propensity for microbial hosts to reside within and feed upon human waste [47], although we cannot be certain of the causitive nature of AMR absence in MG1. It is also important to consider that these findings are all based on in silico genome exploration and gene orthology detection. Accordingly, these data are a first-pass examination of the MG1 isolate and ultimately require the addition of future experiments using in vivo infectious model systems to confirm *bona fide* virulent tendencies of the isolate.

### 4.4. MG1-Specific Detection in Canada Goose Fecal Samples

Positive detection of MG1 in Canada geese fecal samples via our designed real-time PCR assay confirmed the pervasive nature of the MG1 isolate within Canada geese. Combined with the original isolation of MG1 in CTW and Mason, Ohio along with the detection of RM12654 from a Canada goose in California (GenBank: GCA_020137585.1) (see Figure 2), these findings highlight the likelihood of widespread presence of MG1 in this avian species and environments frequented by them. Unfortunately, because of sample degradation, we were unable to use our assay to directly quantify MG1 content within the Banklick Creek sites from which the isolate was initially detected. The real-time PCR assay we developed will be a critical tool in addressing this and other MG1-related questions in future work.

## 5. Summary and Conclusions

We have detected a novel *Campylobacter* isolate, MG1, that was isolated from Canada goose fecal samples. We provided a draft genome assembly and genome overview with a particular emphasis on genetic signatures most often associated with virulence and disease, vis à vis a risk to human and animal health. We have also generated specific primer and probe combinations that can be used for tracking Canada goose fecal contamination. Using these assays, we confirmed the prevalence of MG1 in Canada goose fecal material. Our clone sequencing also showed a high occurrence of MG1-related isolates across a wetland ecosystem frequented by geese. Our findings emphasize the role that waterfowl, specifically Canada geese, may play in acting as a distribution vector for *Campylobacter* species including MG1. In this study, we adopted a selective enrichment approach to identify thermotolerant *Campylobacter* isolates derived from birds that are capable of thriving at human body temperature (37 °C). This allowed us to assess their zoonotic potential and the likelihood of cross-species infection in humans. Moreover, our analysis revealed the presence of multiple MG1 virulence factors, which have been linked to virulence among *Campylobacter* spp. These findings suggest that MG1 may pose a significant threat to human health in the future. Through our MG1-specific assay, future work should focus on examining the prevalence and distribution of this isolate in the environment and, if cases should arise, in human clinical isolates. Combined, these findings can be used to better understand the spread, occurrence, and microbial sources of *Campylobacter* species present in public water sources, and the risk these organisms pose to public health. The identification of this new *Campylobacter* isolate, MG1, in waterbodies used for source drinking water further highlights the importance of using a One Health approach to consider environmental sources and wild animals as another potential reservoir for human-infectious *Campylobacter* spp., and not just anthropogenic or animal/livestock used for human consumption as sources.

## Figures and Tables

**Figure 1 microorganisms-11-00648-f001:**
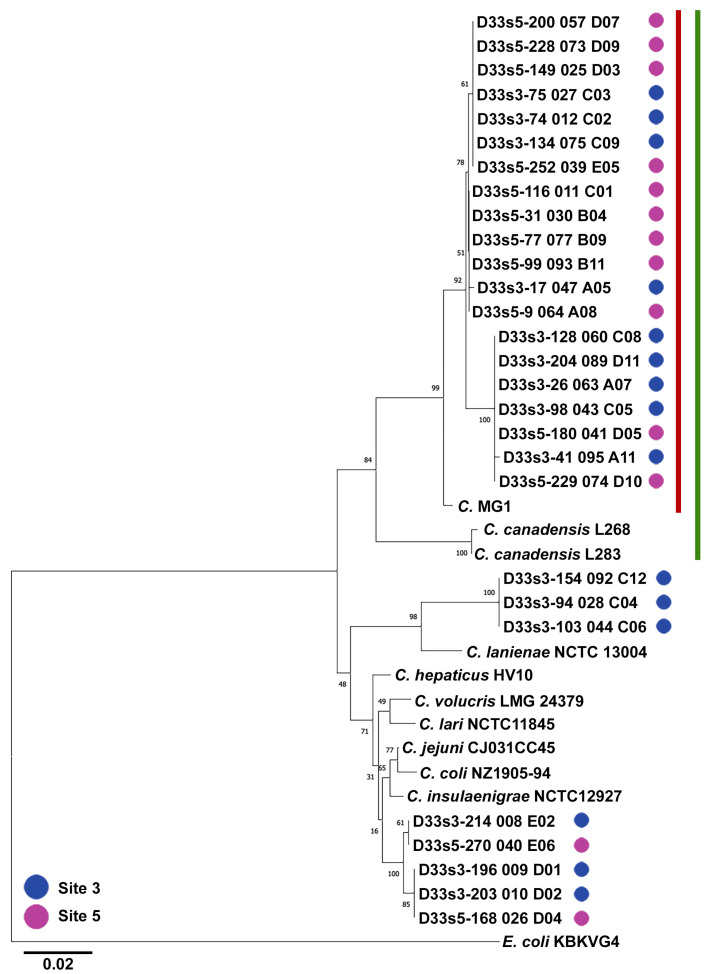
Phylogenetic analysis of *Campylobacter* 16s sequences isolated from Banklick Creek Wetlands site 3 (blue dots) and 5 (purple dots). The neighbor-joining phylogenetic tree is based on nucleotide alignments of 38 sequences composed of 661 nucleotide positions. The optimal tree is shown and drawn to scale in units of the number of base substitutions per site (scale bar = 0.02 substitutions). The percentage of replicate trees in which the associated taxa clustered together in the bootstrap test (500 replicates) are shown next to the branches. *E. coli* (strain KBKVG4) was used to root the tree. The majority of sequenced isolates clustered within a clade containing MG1 (red vertical line). The closest outgroup for the MG1-containing clade was occupied by *C. canadensis* (green vertical line).

**Figure 2 microorganisms-11-00648-f002:**
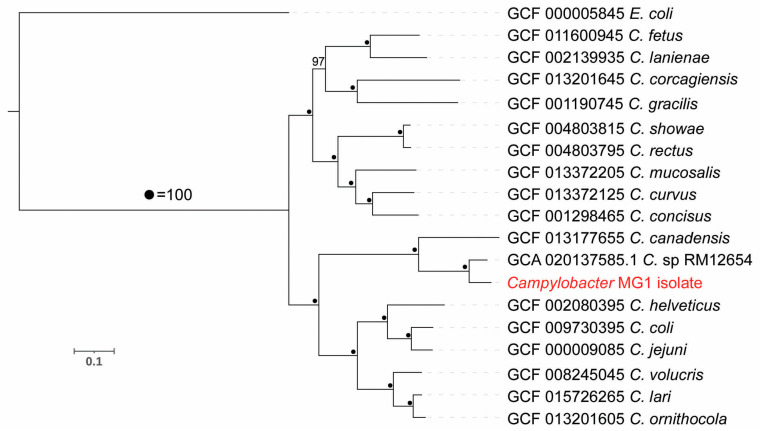
Phylogenomic analysis of the MG1 isolate. The phylogenomic tree was generated using 118 possible single copy genes specific to Proteobacteria and generated via GToTree. *Escherichia coli* (*E. coli*—GCF_000005845) was used to root the tree. Nodes are annotated with bootstrap support (dot = 100% agreement). The scale bar indicates substitutions per site.

**Table 1 microorganisms-11-00648-t001:** MG1 genome assembly statistics.

Coverage (x)	No. of Contigs	Assembly Size (bp)	Contig N50 (bp)	G+C Content (%)	Gene Annotation Data (no.)			
					Genes	CDS	rRNAs	tRNAs
914	92	1,900,729	168,488	26.75	1908	1866	3 (2 *)	39

* two rRNAs after manual curation.

**Table 2 microorganisms-11-00648-t002:** Presumptive MG1 virulence genes identified by BLAST analysis of protein sequences. Query sequences are predominantly from *C. jejuni* (see Materials and Methods).

Query ID	Gene Description	Category	MG1 Best Hit BLAST Results		
			MG1 prokka_aa * ID	Identity (%)	e-Value
YP_002344858.1	outer membrane fibronectin-binding (*cadF*)	Adhesion	PROKKA_01054	41.4	2.48 × 10^−74^
YP_002344378.1	surface-exposed lipoprotein (*jlpA*)	Adhesion	PROKKA_01720	26.7	1.79 × 10^−18^
YP_002344652.1	two-component regulator (*racR*)	Adhesion	PROKKA_01566	65.5	2.56 × 10^−109^
YP_002344653.1	sensor histidine kinase (*racS*)	Adhesion	PROKKA_01567	34.9	1.30 × 10^−56^
YP_002344319.1	bifunctional adhesin/ABC transporter (*peb1A*)	Adhesion	PROKKA_01224	28.6	6.41 × 10^−21^
YP_002344320.1	amino acid ABC transporter ATP-binding protein (*pebC*)	Adhesion	PROKKA_01223	52.5	3.95 × 10^−90^
YP_002344743.1	enterochelin uptake substrate-binding protein (*ceuE*)	Invasion	No significant hits		
YP_002344312.1	invasion antigen B (*ciaB*)	Invasion	PROKKA_01528	40.0	2.92 × 10^−126^
YP_002345016.1	ABC transporter ATP-binding protein (*iamA*)	Invasion	PROKKA_00115	37.0	2.65 × 10^−45^
YP_002343524.1	flagellar motor switch protein (*fliM*)	Invasion	PROKKA_01601	27.5	2.43 × 10^−23^
YP_002343523.1	flagellar motor switch protein (*fliY*)	Invasion	PROKKA_01602	28.6	5.29 × 10^−23^
YP_002344200.1	sensor histidine kinase (*flgS*)	Invasion	PROKKA_01698	48.0	4.48 × 10^−26^
YP_002345095.1	flagellar hook protein (*flgE*)	Invasion	No significant hits		
YP_002344726.1	flagellin B (*flaB*)	Invasion	No significant hits		
YP_002343773.1	flagellar biosynthesis protein (*flhB*)	Invasion	No significant hits		
YP_002343959.1	flagellar basal body rod protein (*flgB*)	Invasion	No significant hits		
YP_002344727.1	flagellin A (*flaA*)	Invasion	No significant hits		
YP_002344282.1	flagellar biosynthesis protein (*flhA*)	Invasion	No significant hits		
YP_002344851.1	type II protein secretion system protein E (*ctsE*)	other	PROKKA_01193	41.4	3.65 × 10^−144^
YP_002344617.1	sensor histidine kinase (Cj1226c)	other	PROKKA_01197	51.2	4.16 × 10^−137^
YP_002344072.1	response regulator domain protein (*cbrR*)	other	PROKKA_00168	41.9	7.56 × 10^−107^
YP_002344977.1	two-component regulator (Cj1608)	other	PROKKA_01490	53.2	8.10 × 10^−104^
YP_002344737.1	fibronectin/fibrinogen-binding (Cj1349c)	other	PROKKA_01122	41.7	3.04 × 10^−98^
YP_002344670.1	fibronectin domain-containing lipoprotein (Cj1279c)	other	PROKKA_00246	39.2	4.35 × 10^−94^
YP_002344872.1	two-component sensor (Cj1492c)	other	PROKKA_01722	38.0	9.64 × 10^−83^
YP_002344532.1	beta-1,3 galactosyltransferase (*wlaN*)	other	PROKKA_00466	53.4	3.78 × 10^−69^
YP_002343492.1	cytochrome C551 peroxidase (*docA*)	other	PROKKA_00798	40.6	1.68 × 10^−63^
YP_002344871.1	two-component regulator (Cj1491c)	other	PROKKA_01721	45.7	1.85 × 10^−54^
YP_002343491.1	MCP-domain signal transduction protein (Cj0019c)	other	PROKKA_01763	37.4	1.34 × 10^−33^
YP_002344318.1	amino acid ABC transporter permease (Cj0920c)	other	PROKKA_01221	29.7	8.74 × 10^−33^
YP_002344289.1	putative sensory transduction transcriptional regulator (Cj0890c)	other	PROKKA_01241	27.7	4.18 × 10^−27^
YP_002344614.1	putative two-component regulator (*dccR*)	other	PROKKA_01196	30.0	2.33 × 10^−24^
WP_002826431.1	virB11	T4SS	PROKKA_01069	28.8	1.5 × 10^−37^
WP_012662267.1	virB10	T4SS	No significant hits		
WP_002834097.1	virB9	T4SS	No significant hits		
WP_012662258.1	virB4	T4SS	PROKKA_00711	23.5	2.12 × 10^−5^
WP_012662270.1	virD4	T4SS	PROKKA_00214	26.5	3.64 × 10^−11^
YP_002343541.1	cytolethal distending toxin B (*cdtB*)	Toxin	No significant hits		
YP_002343539.1	cytolethal distending toxin C (*cdtC*)	Toxin	No significant hits		
YP_002343540.1	cytolethal distending toxin A (*cdtA*)	Toxin	No significant hits		

* prokka_aa are the prokka annotated amino acid sequences from MG1.

**Table 3 microorganisms-11-00648-t003:** Presumptive MG1 group 1 lipooligosaccharide loci identified by BLAST analysis of protein sequences.

*C. jejuni* NCTC11168		MG1 Best Hit BLAST Result		
Query ID	Name	MG1 prokka_aa * ID	Identity (%)	e-Value
YP_002344533.1	*cstIII*	PROKKA_00461	36.7	7.89 × 10^−40^
YP_002344534.1	*neuB1*	PROKKA_00462	64.3	9.39 × 10^−154^
YP_002344535.1	*neuC1*	PROKKA_00463	49.6	4.04 × 10^−117^
YP_002344536.1	*neuA1*	PROKKA_00464	39.6	6.58 × 10^−33^

* prokka_aa are the prokka annotated amino acid sequences from MG1.

**Table 4 microorganisms-11-00648-t004:** Presumptive MG1 capsular polysaccharide loci identified by BLAST analysis of protein sequences.

*C. jejuni* NCTC11168		MG1 Best Hit BLAST Results			
Query ID	Gene Name *	MG1 prokka_aa ** ID	Syntenous ***	Identity (%)	e-Value
YP_002344796.1	*kpsS*	PROKKA_00180	•	44.5	1.11 × 10^−109^
YP_002344797.1	*kpsC*	PROKKA_00172	•	54.4	0
YP_002344798.1	*cysC*	No significant hits			
YP_002344799.1	—	PROKKA_00458		33.8	3.12 × 10^−8^
YP_002344800.1	—	PROKKA_00080		23.0	4.74 × 10^−7^
YP_002344801.1	—	No significant hits			
YP_002344802.1	—	PROKKA_00794		32.6	6.31 × 10^−9^
YP_002344803.1	—	No significant hits			
YP_002344804.1	—	No significant hits			
YP_002344805.1	—	No significant hits			
YP_002344806.1	*hddC*	PROKKA_01216		29.0	2.93 × 10^−10^
YP_002344807.1	*gmhA2*	PROKKA_00892		49.1	2.72 × 10^−47^
YP_002344808.1	*hddA*	No significant hits			
YP_002344809.1	—	No significant hits			
YP_002344810.1	—	PROKKA_01532		26.7	4.69 × 10^−17^
YP_002344811.1	*fcl*	No significant hits			
YP_002344812.1	—	No significant hits			
YP_002344813.1	*rfbC*	No significant hits			
YP_002344814.1	*hddC*	No significant hits			
YP_002344815.1	—	No significant hits			
YP_002344816.1	—	No significant hits			
YP_002344817.1	—	PROKKA_00176	•	36.0	5.90 × 10^−23^
YP_002344818.1	—	No significant hits			
YP_002344819.1	—	PROKKA_00189	•	25.9	3.38 × 10^−11^
YP_002344820.1	—	PROKKA_00189	•	30.6	5.00 × 10^−20^
YP_002344821.1	—	PROKKA_00176	•	34.2	8.19 × 10^−22^
YP_002344822.1	*glf*	PROKKA_00175	•	36.8	2.25 × 10^−78^
YP_002344823.1	—	PROKKA_00173	•	33.5	7.61 × 10^−52^
YP_002344824.1	*kfiD*	No significant hits			
YP_002344825.1	—	No significant hits			
YP_002344826.1	*kpsF*	PROKKA_00337		42.0	3.12 × 10^−69^
YP_002344827.1	*kpsD*	PROKKA_00184	•	55.1	0
YP_002344828.1	*kpsE*	PROKKA_00183	•	43.3	5.24 × 10^−87^
YP_002344829.1	*kpsT*	PROKKA_00182	•	63.6	2.84 × 10^−110^
YP_002344830.1	*kpsM*	PROKKA_00181	•	43.9	3.50 × 10^−67^

* Dash symbol indicates genes are unnamed, ** prokka_aa are the prokka annotated amino acid sequences from MG1, *** hits are located along the same contig. Dot (•) confirms synteny.

## Data Availability

All raw genome sequencing data have been deposited in the NCBI sequence read archive under accession number PRJNA875047. The assembled genome sequence has been deposited at DDBJ/ENA/GenBank under accession number JANYME000000000.

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
