# Peer review of "Genomic Characterization and Wetland Occurrence of a Novel *Campylobacter* Isolate from Canada Geese"

_microorganisms, 2023, doi:10.3390/microorganisms11030648_

Round 1
Reviewer 1 Report
-
Author Response
Reviewer 1 has not provided any specific comments. We hope our edits in response to other reviewers will alleviate any concerns they had with the content of our manuscript.

Reviewer 2 Report
Dear authors,
although the idea that guided yours work was a good idea: the experimentation is inadequate. The number of samples is small, the sampling itself did not have a precise design, the isolation temperature of Campylobacter from birds is 42°C (choosing 37°C you probably have lost other Campylobacter like species) and the conclusions seem inconsistent on a One Health viewpoint.
I think it would be best to review and repeat the sampling and isolation before publishing the data.
Best Regards.
Author Response
Dear authors,
although the idea that guided yours work was a good idea: the experimentation is inadequate. The number of samples is small, the sampling itself did not have a precise design, the isolation temperature of Campylobacter from birds is 42°C (choosing 37°C you probably have lost other Campylobacter like species) and the conclusions seem inconsistent on a One Health viewpoint.
I think it would be best to review and repeat the sampling and isolation before publishing the data.
Best Regards.
We thank the reviewer for their constructive criticism. Although our sampling approach had some limitations, we would argue that, despite the reviewer’s comments, the information currently presented in this manuscript already represents a significant contribution to the field. This study was not only successful in revealing a new and apparently prominent clade of Campylobacter canadensis-like strains in urban wetlands but also in genetically characterizing a novel Campylobacter isolate that may be representative of this clade. This study also provides the research community with valuable new genomic resources for this strain, including a preliminary genomics-based overview of its zoonotic potential. Additionally, we have developed new real-time PCR resources for future analyses, further enhancing the utility of our findings. Moreover, as stated in the discussion section of this manuscript, our previous study (McMinn et al., 2019) showed statistically significant correlation with bird occupancy and the presence of Campylobacter spp. (including, we now know, the MG1 isolate) in the CTW using qPCR, indicating that Canada geese pose a significant source of fecal contamination that has impacted the water quality going through the CTW.
Regarding our isolation protocol – this protocol was developed according to Hsieh et al. (2018). Hsieh et al. found that incubation at 37C was the optimal temperature for the recovery and the growth of most Campylobacter species. Our isolation was designed to obtain Campylobacter canadensis-like species and not focused on the isolation of other less dominant Campylobacter species. Based on this, we determined that 37C was an appropriate temperature to isolate our target – and indeed we found the exact isolate that our initial Banklick Creek genetic analysis (16s-based) suggested was present throughout the Banklick Creek CTW. These additional details have now been incorporated in the materials and methods section of the revised manuscript (pg. 3) and reads, “The plates were incubated at 37°C, which was found to be the optimal temperature for the recovery and growth of most Campylobacter species (Hsieh et al., 2018), for 30 minutes.”
Hsieh YH, Simpson S, Kerdahi K, Sulaiman IM. A Comparative Evaluation Study of Growth Conditions for Culturing the Isolates of Campylobacter spp. Curr Microbiol. 2018 Jan;75(1):71-78. doi: 10.1007/s00284-017-1351-6. Epub 2017 Sep 8. PMID: 28887647.
With regards to the One Health initiative, we have modified the summary and conclusion section of the revised manuscript to describe the importance of environmental reservoirs and wild animals as the third major component that also requires serious attention when it comes to the One Health Initiative. The passage (pg. 9) now reads, “The identification of this new Campylobacter isolate, MG1, in waterbodies used for source drinking water further highlights the importance of using a One Health approach to consider environmental sources and wild animals as another potential reservoir for human-infectious Campylobacter spp., and not just anthropogenic or animal/livestock used for human consumption as sources.”

Reviewer 3 Report
The manuscript describes the presence of Campylobacter in the population of wild geese and describes a novel isolate . The manuscript is clear and well written and there are only minor details that should be considering before publication.
pag. 1 - Escherichia coli instead of E. coli
Pag 2- Last paragraph of Introduction is not very clear and possibly would be better to include in Material and Methods and state the objetives of the work at the end of ontroduction
pag 2 - First and second phrase of materials and methods might be irrelevant. Consider deleting it.
Page 5- “…C. canadensis, a
novel Campylobacter isolate obtained from cloacal swabs obtained from captive adult
whooping cranes (Grus americana).! Needs reference.
Page 7 - Regarding the phrases: “The absence of AMR and toxin-producing genes present in MG1 lessens the danger
of human and animal transmission or health effects. The lack of AMR genes is a promising
finding given the increasing prevalence of resistance to common antibiotics found among
Campylobacter spp. frequently found in wild birds (Ahmed & Gulhan, 2022; Du et al.,
2019). These findings also highlight the lack of crossover between MG1, MG1 avian
vectors, and human/human-related waste, as AMR prevalence is associated with the
propensity for microbial hosts to reside within and feed upon human waste (Gil & Hird,
2022).” In my opinion the lack of AMR and toxin producing genes in Campylobacter MG1 do not imply that the isolate is not circulation between hosts and the environment. Please rephrase.
The supplementary material was not provided.
Author Response
Reviewer 3
The manuscript describes the presence of Campylobacter in the population of wild geese and describes a novel isolate . The manuscript is clear and well written and there are only minor details that should be considering before publication.
pag. 1 - Escherichia coli instead of E. coli
We thank the reviewer for catching this mistake and we have adjusted the wording as suggested.
Pag 2- Last paragraph of Introduction is not very clear and possibly would be better to include in Material and Methods and state the objetives of the work at the end of ontroduction
We agree with the reviewer that the last paragraph of the introduction was not as clear as is needed. We kept the paragraph but made substantial changes that we hope will clarify the experimental outline of our study and state the objectives and impact of our work.
pag 2 - First and second phrase of materials and methods might be irrelevant. Consider deleting it.
We have made some changes to the text in these sections of the materials and methods. We did opt to keep the sections, as we feel it is necessary for readers to understand why our fecal isolation, from which MG1 was isolated, came from a separate source (Mason, Ohio) compared to the initial experimental location (Banklick Creek CTW).
Page 5- “…C. canadensis, a novel Campylobacter isolate obtained from cloacal swabs obtained from captive adult whooping cranes (Grus americana).! Needs reference.
We have added a reference (Inglis et al., 2007) for this statement.
Inglis GD, Hoar BM, Whiteside DP, Morck DW. (2007). Campylobacter canadensis sp. nov., from captive whooping cranes in Canada. Int J Syst Evol Microbiol. 57(11), 2636-2644.
Page 7 - Regarding the phrases: “The absence of AMR and toxin-producing genes present in MG1 lessens the danger of human and animal transmission or health effects. The lack of AMR genes is a promising finding given the increasing prevalence of resistance to common antibiotics found among Campylobacter spp. frequently found in wild birds (Ahmed & Gulhan, 2022; Du et al., 2019). These findings also highlight the lack of crossover between MG1, MG1 avian vectors, and human/human-related waste, as AMR prevalence is associated with the propensity for microbial hosts to reside within and feed upon human waste (Gil & Hird, 2022).” In my opinion the lack of AMR and toxin producing genes in Campylobacter MG1 do not imply that the isolate is not circulation between hosts and the environment. Please rephrase.
We have adjusted the wording in the described paragraph to soften our conclusions regarding how association of MG1 and MG1-hosts with human waste could be directly causative on the prevalence (or lack thereof) of anti-microbial resistance (AMR). With the adjusted wording, we opted to keep the discussion of this point as we feel it is a meaningful discussion topic for the new MG1 isolate.

Reviewer 4 Report
The study describes the finding and characterisation of a novel isolate of Campylobacter, of potential major public health importance. Although the work is important, methodologically correct and well written, there are several modifications that have to be done, whereas the structure needs re organisation. Particularly I suggest to divide the Results and Discussion in separate sections and further Discussion to subsections. Also the developed molecular technique needs more details, for instance in how many samples tested and how many sequences were used for the design of the primers
Some more detailed comments:
Why the authors used clone sequencing? I do not like very much the term. If the authors mean genetic analysis, please state so
Also, “better identify” is not a correct term. Either we can identify, that means determine the name of the species, or better analyse. Please rephrase
In the abstract please also add information regarding the Banklick Creek wetland, at least the country. Also provide more details in the first time mentioned in the Introduction.
The encode Campylobacter cytolethal distending toxin is responsible for pathogenicity in all hosts? Same as flagellin. Please provide this detail in both parts
In the scope paragraph, please explain better why the Mason city park (40 miles away from the CTW) was chosen for collection of extra samples.
In “MG1 16s rRNA Primer Design”, please refer how many sequences were obtained from each species for the development of the technique. It is very important to ensure there is no genetic diversity within species at the targeted SNPa
I am not sure Quantitative PCR is the appropriate term. It is not really a qPCR since there is no control, standard curve and quantification. An other term should be used, e.g. real time PCR or species specific real time PCR
Author Response
Reviewer 4
The study describes the finding and characterisation of a novel isolate of Campylobacter, of potential major public health importance. Although the work is important, methodologically correct and well written, there are several modifications that have to be done, whereas the structure needs re organisation. Particularly I suggest to divide the Results and Discussion in separate sections and further Discussion to subsections. Also the developed molecular technique needs more details, for instance in how many samples tested and how many sequences were used for the design of the primers
We thank the reviewer for the careful consideration of our manuscript and the suggestion to divide our Results and Discussion. As the reviewer requested, we have separated these sections and we agree it helps clarify our findings. Regarding the samples tested and sequences for primer design please see our response below.
Some more detailed comments:
Why the authors used clone sequencing? I do not like very much the term. If the authors mean genetic analysis, please state so
We have adjusted the use of ‘clone sequencing’ to ‘genetic analysis’ where appropriate.
Also, “better identify” is not a correct term. Either we can identify, that means determine the name of the species, or better analyse. Please rephrase
We adjusted this phrase by removing the word “better” in the abstract.
In the abstract please also add information regarding the Banklick Creek wetland, at least the country. Also provide more details in the first time mentioned in the Introduction.
We added clarification and location information in both the abstract and introduction. We included a section providing additional details about Banklick Creek that reads, “Sanitation District No. 1 (SD1) in Fort Wright, Kentucky (USA) installed a constructed treatment wetland (CTW) in 2011 to divert and treat a portion of the Banklick Creek, a watershed regularly impacted by combined sewer overflows (CSOs) and sanitary sewer overflows (SSOs) in the area. CTWs can be an effective means to passively treat water by removing harmful chemical and microbial contaminates, but since they closely mimic natural wetlands, can attract native wildlife such as waterfowl.”
The encode Campylobacter cytolethal distending toxin is responsible for pathogenicity in all hosts? Same as flagellin. Please provide this detail in both parts
Campylobacter cytolethal distending toxin (CDT) and flagellin are both important virulence factors in Campylobacter species and have been implicated in the pathogenicity of this bacterium in humans and animals. However, the exact role of these virulence factors in pathogenicity can vary depending on the host. We tried to clarify these points in the introduction of the manuscript by adding a portion of a sentence that states, “as critical to their human and animal pathogenicity”.
In the scope paragraph, please explain better why the Mason city park (40 miles away from the CTW) was chosen for collection of extra samples.
We thank the reviewer for pushing us on this point. We clarified the writing in both the introduction and methods. In the writing we tried to emphasize that initially we attempted to sample the Banklick Creek CTW for Canada goose feces for isolation of the unknown Campylobacter isolate. Unfortunately, these samples were gathered after some time, and because of the age of the feces we could not successfully culture any Campylobacter spp. from the samples. Instead, we chose a more local location (Mason, Ohio – about 40 miles away from the CTW) where we could actively monitor goose activity and collect extremely fresh samples. This effort allowed us to successfully culture Campylobacter spp. including, for our study, MG1. We altered the text (especially in the last paragraph of the introduction) to clarify the use of the Mason park for fecal collection.
In “MG1 16s rRNA Primer Design”, please refer how many sequences were obtained from each species for the development of the technique. It is very important to ensure there is no genetic diversity within species at the targeted SNPa
We have added an additional supplementary figure (Fig. S2) which shows the multiple sequence alignment of the region where we designed our primers. Additionally, using BLAST, each representative sequence from each species was inspected against publicly available sequences from the respective species to confirm homology within the region across strains. We added clarification to the methods to highlight these points.
I am not sure Quantitative PCR is the appropriate term. It is not really a qPCR since there is no control, standard curve and quantification. An other term should be used, e.g. real time PCR or species specific real time PCR
We agree with the reviewer and have changed all instances of ‘qPCR’ to ‘real-time PCR’.

Round 2
Reviewer 2 Report
Dear authors, I really appreciated your response. I carefully reread your article using a different, more perspective point of view and appreciated its validity. I think there are still some points for further investigation, especially regarding the insulation temperature. In fact by using 37°C you have put thermotolerant Campylobacter, which at the body temperature of birds find their natural habitat, in suboptimal growth conditions. So MG1 would probably have had no relevance when evaluated together with the other Campylobacters of human health concern. The problem have been the "Key of interpretation" of the data.That said, it remains an interesting finding in terms of biodiversity and knowledge of the microbial population of wild birds living in the vicinity of waters usable for human consumption. So, actually, I would avoid to pose MG1 as an element of potential pathogenicity to humans, and talk about it in terms of "prevalence", but in terms of "occurrence". If you would talk about prevalence, I would recommend to do a metagenome assessment of Canada goose feces to evaluate its actual prevalence, taking advantage of the in-depth characterization that you have done.
Best regards.
Reviewer 4 Report
I am satisfied with the modified version of the manuscript and hence I suggest publication in its current form
Author Response
We thank the reviewer for their positive words and for re-examining our manuscript.